# HLA Class I Expression Is Associated with DNA Damage and Immune Cell Infiltration into Dysplastic and Neoplastic Lesions in Ulcerative Colitis

**DOI:** 10.3390/ijms241713648

**Published:** 2023-09-04

**Authors:** Haruka Okami, Naoya Ozawa, Makoto Sohda, Takehiko Yokobori, Katsuya Osone, Bilguun Erkhem-Ochir, Gendensuren Dorjkhorloo, Takuya Shiraishi, Takuhisa Okada, Akihiko Sano, Makoto Sakai, Tatsuya Miyazaki, Hiroomi Ogawa, Takashi Yao, Takahiro Oike, Hiro Sato, Ken Shirabe, Atsushi Shibata, Hiroshi Saeki

**Affiliations:** 1Department of General Surgical Science, Graduate School of Medicine, Gunma University, Maebashi 371-8510, Japan; m2220010@gunma-u.ac.jp (H.O.); a050055asahi@gmail.com (N.O.); okatsuya@gunma-u.ac.jp (K.O.); m2220603@gunma-u.ac.jp (G.D.); whityshiro@gmail.com (T.S.); t.okd@gunma-u.ac.jp (T.O.); ak_sano@outlook.jp (A.S.); maksakai@gunma-u.ac.jp (M.S.); hiroomio@gunma-u.ac.jp (H.O.); kshirabe@gunma-u.ac.jp (K.S.); h-saeki@gunma-u.ac.jp (H.S.); 2Division of Integrated Oncology Research, Gunma University, Initiative for Advanced Research (GIAR), Maebashi 371-8511, Japan; bilguun.e@gunma-u.ac.jp; 3Department of Surgery Japanese Red Cross Maebashi Hospital, Maebashi 371-0811, Japan; tatsuyamiyazaki4126@gmail.com; 4Department of Human Pathology, Graduate School of Medicine, Juntendo University, Bunkyo-ku 113-8431, Japan; tyao@juntendo.ac.jp; 5Department of Radiation Oncology, Graduate School of Medicine, Gunma University, Maebashi 371-8510, Japan; oiketakahiro@gunma-u.ac.jp (T.O.); hiro.sato@gunma-u.ac.jp (H.S.); 6Division of Molecular Oncological Pharmacy, Faculty of Pharmacy, Keio University, Minato-ku 108-8345, Japan; shibata.at@keio.jp

**Keywords:** human leukocyte antigen, DNA damage, colitic cancer, immune cells

## Abstract

Human leukocyte antigen class I (HLA-I) is considered a genetic pathogen for ulcerative colitis (UC). This study aimed to investigate the significance of DNA damage and HLA-I expression in infiltrating immune cells and immune checkpoint protein PD-L1 expression in dysplasia/colitic cancer (CC) and sporadic colorectal cancer (SCRC). We performed immunohistochemical staining for HLA-I, PD-L1, γH2AX (DNA damage marker), and immune cell markers such as CD8, FOXP3, CD68, and CD163 (in surgically resected specimens from 17 SCRC patients with 12 adjacent normal mucosa (NM) and 9 UC patients with 18 dysplasia/CC tumors. The ratio of membrane HLA-I-positive epithelial cells in UC and dysplasia/CC tissues was significantly higher than that in NM and SCRC. High HLA-I expression in dysplasia/CC was associated with high positivity of γH2AX and PD-L1 expression compared to SCRC. The infiltration of CD8-positive T cells and CD68-positive macrophages in HLA-I-high dysplasia/CC was significantly higher than in UC and SCRC. Dysplasia/CC specimens with DNA damage exhibited high levels of HLA-I-positive epithelial cells with high CD8- and CD68-positive immune cell infiltration compared to UC and SCRC specimens. Targeting DNA damage in UC may regulate immune cell infiltration, immune checkpoint proteins, and carcinogenesis by modulating DNA damage-induced HLA-I antigen presentation.

## 1. Introduction

Non-steroidal anti-inflammatory drugs, such as aspirin, have been reported to prevent sporadic colorectal cancer (SCRC) by suppressing chronic inflammation in the colorectal mucosa [1,2,3]. However, chronic colon inflammation due to autoimmune responses could promote ulcerative colitis (UC) and rare colitis-associated dysplasia/colitic cancer (CC) via aggressive infiltration of inflammatory lymphocytes and macrophages [4,5]. The human leukocyte antigen (HLA) molecule is implicated in the pathogenesis of autoimmune chronic inflammation of the colon. The highly polymorphic classical class I and II HLA genes located at chromosome 6p21.3 are important for lymphocyte and immunoregulatory functions in patients with inflammatory bowel diseases, including UC [6]. Many researchers have reported the importance of HLA class II genotypes such as HLA-DRB1*0103, HLA-DRB1*1502, and HLA-DRB1*04 in relation to UC susceptibility. On the other hand, the significant expression of HLA class I (HLA-I) on the cell surface, which is important for presenting antigens to cytotoxic T cells, has been reported in patients with several cancer types [7,8,9,10]. Regarding SCRC, HLA-I expression was lost or altered in 30–73% of patients with tumors [11,12]: suggesting that lack of HLA-I expression may cause both initial resistance and secondary immune escape after existing immune checkpoint inhibitors due to cytotoxic T cell activation [13,14,15]. However, few studies have assessed the significance of HLA-I expression in patients with relatively rare UC and dysplasia/CC. Therefore, we focused on assessing the relationship between immune cell infiltration and HLA-I expression, not HLA-class II, on the epithelial cells in SCRC and dysplasia/CC tissues in this study.

Uchihara et al. have recently described a novel regulation mechanism of HLA-I expression by which antigen transport via TAP1/2 could promote the presentation of HLA-I to the plasma membrane regardless of DNA damage-inducing methods, such as radiation or anticancer drugs [16]. DNA damage-induced HLA-I presentation could be predicted to activate signaling downstream of T-cell immunity. In contrast, Sato et al. reported that expression of the immune checkpoint protein PD-L1 is upregulated by the DNA damage response [17]. These reports indicated that DNA damage could induce two opposing events: inflammatory activation via antigen presentation by HLA-I and immune tolerance by induction of the immune checkpoint protein PD-L1. Concerning the relationship among DNA damage, PD-L1 expression, and immune cell infiltration in dysplasia/CC tissues, our group reported that CD8-positive cytotoxic T-cell infiltration was higher in PD-L1 expressing dysplasia/CC tissues with DNA damage than in those with SCRC without DNA damage [18]. However, the effects of HLA-I on immune cell infiltration in dysplasia/CC tissues with DNA damage and PD-L1 expression have not been thoroughly evaluated. This study aimed to compare the association between HLA-I expression levels and immune cell infiltration in dysplasia/CC tissues, which we previously reported to accumulate DNA damage and PD-L1, with tissues from SCRC samples.

## 2. Results

### 2.1. Evaluation of Membrane HLA-I Expressing Epithelial Cells in Tissues of Patients with SCRC and Dysplasia/Colitic Cancer

We performed immunohistochemistry to assess HLA-I expression in clinical specimens from 12 NM, 17 SCRC, 9 UC, and 18 dysplasia/CC patients using specific antibodies against HLA-Class I A, B, and C. HLA-I expression was predominantly detected in the membrane of the epithelial cells in tissues from UC and CC samples (Figure 1A). We evaluated the average number of membrane HLA-I-positive epithelial cells for further analysis, which was significantly higher in UC (n = 9) and dysplasia/CC (n = 18) specimens than those in NM and SCRC specimens (Figure 1B).

### 2.2. Relationship between HLA-I Expression, DNA Damage, and PD-L1 Expression in SCRC and Dysplasia/CC

HLA-I-stained dysplasia/CC specimens were divided into low and high groups according to the median number of HLA-I-positive dysplasia/CC cells (cutoff value: 21.5). Thus, dysplasia/CC specimens were categorized HLA-I group and low HLA-I group (Table 1). According to the HLA-I cutoff value, all 12 NM and 17 SCRC specimens were evaluated as being in the low HLA-I group (Table 2). The positive ratio of the DNA damage marker γH2AX in dysplasia/CC (100%, 18/18) was higher than that in SCRC without HLA-I expression (52.9%, 9/17). Moreover, the positive ratio of the representative immune checkpoint protein PD-L1 to the representative HLA-I inducible factor interferon regulatory factor-1 (IRF-1) in dysplasia/CC (PD-L1: 100%, 18/18, and IRF-1: 100%, 18/18, respectively) was higher than that in SCRC without HLA-I expression (PD-L1: 35.3%, 6/17, and IRF-1: 41.2%, 7/17, respectively).

### 2.3. Immune Cell Infiltration in Tissues of SCRC, UC, and Dysplasia/CC with Low and High HLA-I

Immunohistochemical tests were performed to evaluate tissue-infiltrating immune cells using immune cell markers such as CD8 (cytotoxic T cell marker), CD68 (M1-like pro-inflammatory macrophage marker), CD163 (M2-like anti-inflammatory macrophage marker), and FOXP3 (regulatory T cell marker) in the clinical specimens of patients with SCRC and patients with UC with dysplasia/CC (Figure 2). The number of infiltrating immune cells with CD8 and CD68 expression in dysplasia/CC with high HLA-I levels was higher than that in UC and SCRC (Figure 3A,C); however, this was not significant in dysplasia/CC with low CD8 infiltrating immune cells. Figure 4 shows that CC cells with high membrane HLA-I expression were associated with higher infiltration of CD8 + T lymphocytes compared to those with low HLA-I expression (Figure 4). In contrast, the number of infiltrating immune cells expressing CD163 and FOXP3 in dysplasia/CC with high HLA-I levels was higher than that in UC (Figure 3B,D). 

## 3. Discussion

In this study, we performed immunohistochemical staining for HLA-I, PD-L1, γH2AX (DNA damage marker), IRF-1 (a representative HLA inducible factor), and immune cell markers such as CD8 (cytotoxic T cell marker), FOXP3 (regulatory T cell marker), CD68 (M1-like pro-inflammatory macrophage marker), and CD163 (M2-like anti-inflammatory macrophage marker) in surgically resected specimens from 17 patients with SCRC with 12 adjacent normal mucosa (NM) and 9 patients with UC with 18 dysplasia/CC tumors. Our results demonstrated a significant increase in the number of HLA-I-positive epithelial cells in UC and dysplasia/CC tissues compared to NM and SCRC tissues. In addition, the high HLA-I expression in dysplasia/CC tissues was associated with high expression of the DNA damage marker (γH2AX) and a representative immune checkpoint protein (PD-L1) compared to SCRC tissues without high HLA-I. Moreover, the infiltration of CD8-positive T cells and CD68-positive macrophages in dysplasia/CC tissues with high HLA-I levels was significantly higher than that in UC and SCRC tissues.

Although the HLA genotype has been reported to be associated with susceptibility, degree of colitis, and extraintestinal complications in patients with UC, the relationship between HLA-I expression levels (but not the HLA genotype) and inflammatory cell infiltration in dysplasia/CC has not been extensively investigate [19,20]. HLA-I-activated CD8-positive cytotoxic T cells secrete IFN-γ, which contributes to the apoptotic induction of tumor cells by promoting HLA-I expression [21]. In addition, the IFN-γ can promote HLA-I gene transcription by activating NF-κB and IRF1, which bind to the promoter region of HLA-I [22]. In the present study, dysplasia/CC had a higher positivity for the DNA damage marker γH2AX and IRF-1 regarding HLA-I expression than SCRC. Moreover, HLA-I-high dysplasia/CC showed significant infiltration of CD8-positive cytotoxic T cells compared to SCRC and UC. These findings suggest that the mechanism of increased HLA-I expression in dysplasia/CC is not only due to DNA damage caused by UC-induced inflammatory cell infiltration but also to the activation of inflammatory cell-derived IFN-γ signaling. We also considered that the increased inflammatory cell infiltration in combination with DNA damage in dysplasia/CC was consistent with the hypothesis of previous reports [16] that DNA damage-induced HLA-I expression contributes to the presentation of immunogenic peptide antigens.

Considering the significance of DNA damage in dysplasia/CC, two therapeutic strategies are required. First, to prevent carcinogenesis, immune cell activation by antigen presentation by HLA-I is considered crucial for tumor elimination. However, DNA damage-induced HLA-I due to chronic inflammation may promote the initiation of dysplasia/CC of the UC colonic mucosa via the enhancement of inflammatory cell infiltration based on the inflammation-dysplasia-carcinoma sequence theory [23,24]. To interrupt the unfavorable inflammatory carcinogenic sequence, anti-inflammatory drugs, such as 5-aminosalicylates and aspirin, which contribute to the control of inflammation in UC and the prevention of carcinogenesis in SCRC, may be promising [25,26,27,28,29,30]. In addition, therapeutic strategies that inhibit the infiltration of inflammatory cells into the colonic mucosa of patients with UC are currently being investigated. Vedolizumab, an anti-α4β7-integrin antibody against integrin α4β7 expressed on inflammatory cells in the colon, and a specific antibody against MAdCAM-1 expressed on vascular endothelial cells in the UC mucosa have been reported to be able to control UC-induced inflammation [31,32,33,34,35,36,37]. Examining whether these therapeutic strategies can prevent the initiation of dysplasia or CC is a significant clinical challenge.

Secondly, regarding advanced CC treatment, previous studies and our data show that rare dysplasia/CC has significantly accumulated DNA damage, HLA-I overexpression, high inflammatory cell infiltration, and high immune checkpoint protein PD-L1 expression compared to typical SCRC. Immune checkpoint inhibitors (ICIs), which have recently attracted considerable attention, are insensitive to cold tumors without inflammatory cell infiltration or immune checkpoint proteins. In contrast, hot tumors with a high degree of inflammatory cell infiltration and accumulation of the immune checkpoint protein PD-L1 have been reported to be sensitive to ICIs [38,39,40]. Therefore, it was suggested that dysplasia/CC with HLA-I upregulation might be highly sensitive to ICIs because it possesses hot tumor features, such as significant DNA damage, high inflammatory cell infiltration, and PD-L1 accumulation. Indeed, the HLA-I genotype has been reported to be associated with prognosis in patients with cancer treated with ICIs, suggesting the importance of HLA-I in ICI sensitivity [41]. In contrast, as ICI has been reported to cause colitis as a side effect, future studies should be conducted to determine whether the degree of colitis as a side effect is worse in patients with HLA-I-high UC and dysplasia/CC with high inflammatory cell infiltration. 

This study has several limitations. A limited number of subjects may have underestimated the significance of DNA damage-induced HLA-I presentation as a biomarker in rare dysplasia/CC samples from patients with UC. As a retrospective study, this could not determine whether DNA damage-induced HLA-I can facilitate inflammation and carcinogenesis in the UC mucosa via the activation of antigen presentation by membrane HLA-I. Moreover, we could not perform in vitro or in vivo functional analyses of the relationship between DNA damage-induced HLA-I and tumor immunity using cell lines derived from patients with UC and dysplasia/CC.

In conclusion, this study revealed that dysplasia/CC specimens with DNA damage exhibited high levels of membrane HLA-I-positive epithelial cells with high CD8- and CD68-positive immune cell infiltration compared to UC and SCRC specimens. Previous in vitro studies have shown that DNA damage can induce HLA-I antigen presentation and immune checkpoint protein PD-L1 expression in tumor immunity [16]; however, these findings have not been validated in rare dysplasia or CC specimens. We clarified the important relationship between DNA damage, HLA-I accumulation, and immune cell infiltration in clinical dysplasia/CC samples. Thus, therapeutic strategies to prevent DNA damage in patients with UC may regulate immune cell infiltration, immune checkpoint protein expression, and carcinogenesis by controlling antigen presentation via DNA damage-induced HLA-I.

## 4. Materials and Methods

### 4.1. Patients and Samples

Nine patients (six males and three females) with UC who underwent surgical resection at Gunma University Hospital and Maebashi Red Cross Hospital between 2000 and 2014 were included in this study. The median age of the patients was 51 years (range 37–72 years). Two patients had two or more tumors, and all high-grade dysplastic and cancerous lesions were evaluated. Patients with low-grade dysplasia were not included in the study. Seventeen patients with SCRC (12 male and 5 female patients) who underwent partial colectomy at Gunma University Hospital between 2000 and 2014 were randomly selected and included in this study. This study conformed to the tenets of the Declaration of Helsinki and was approved by the Institutional Review Board for Clinical Research of Gunma University Hospital (approval number: HS2023-027). Patient consent was obtained by using the opt-out method. Table 1 and Table 2 summarize the patient information. For accurate pathological diagnosis of dysplastic and cancerous lesions in patients with UC, all sections were evaluated by a specialized pathologist, Prof. Yao T (Department of Human Pathology, Juntendo University Graduate School of Medicine).

### 4.2. Immunohistochemical Staining

Paraffin-embedded blocks were cut into four µm-thick sections and mounted on glass slides. Sections were deparaffinized in xylene and dehydrated in alcohol. The endogenous peroxidase activity was inhibited by incubation with 0.3% H_2_O_2_/methanol for 30 min at room temperature. After rehydration through a graded series of ethanol treatments, antigen retrieval was performed using an Immunosaver (Nisshin EM, Tokyo, Japan) at 98–100 °C for 45 min, and PD-L1 was retrieved using Universal HIER antigen retrieval reagent (Abcam, Cambridge, UK, ab208572) at 120 °C for 20 min in an autoclave. Non-specific binding sites were blocked by incubation with Protein Block Serum-Free (Dako, Carpinteria, CA, USA) for 30 min. Next, the sections from the clinical samples were incubated overnight at 4 °C with primary antibodies against HLA Class I-A, B, C (Hokudo, Sapporo, Japan, EMR8-5, mouse mAb, 1:400 dilution), CD8 (DAKO, C8/144B, mouse mAb, 1:100 dilution), CD68 (Abcam, ab955, mouse mAb, 1:100 dilution), CD163 (Cell Signaling Technology, Beverly, MA, USA, D6U1J, rabbit mAb, 1:500 dilution), FOXP3 (Abcam, ab22510, mouse mAb, 1:100 dilution), anti-PD-L1 (Abcam, ab205921, 28-8, Rabbit mAb, 1:200 dilution), anti-γH2AX (Abcam, ab26350, 9F3, mouse mAb, 1:200), anti-IRF-1 (Abcam, ab186384, Rabbit mAb, 1:300). The Histofine Simple Stain MAX-PO (Multi) Kit (Nichirei, Tokyo, Japan) was used with the secondary antibody at room temperature for 30 min. The chromogen 3,3′-diaminobenzidine tetrahydrochloride was applied as a 0.02% solution containing 0.005% H_2_O_2_ in 50 mM ammonium acetate-citrate acid buffer (pH 6.0). Nuclear counterstaining was performed using Mayer’s hematoxylin solution. Negative controls for immunohistochemical staining were prepared by replacing the primary antibodies with phosphate-buffered saline in 0.1% bovine serum albumin, confirming a lack of staining.

### 4.3. Immunohistochemical Evaluation

We counted the number of HLA-I-positive epithelial cells in the tissues and evaluated the average number of cells in five high-power fields of view at high-powered fields (magnification, ×400). HLA-I-stained samples were divided into low and high groups according to the median HLA-I-positive cell number in the dysplasia/CC samples (cutoff value: 21.5). We considered tumor and non-cancerous cells that displayed membrane PD-L1 staining as positive when at least 1% of the cells were stained [42]. The staining intensity for γH2AX was scored as follows: 0 (no staining); 1+ (weak staining); 2+ (moderate staining); and 3+ (strong staining). The percentage of nuclear-stained cells was determined by examining the three sections with the highest staining intensity. The proportion of nuclear γH2AX staining was scored as follows:0 (no staining); 1+, 1–25%; 2+, 26–50%; and 3+, 51–100%. The final score was defined as the ratio of the score multiplied by the intensity score (0, 1+, 2+, 3+, 4+, 6+, or 9+). Nuclear immunoreactivity of γH2AX was scored as 0–4+ and 6–9+, which were defined as low and high nuclear expression, respectively. Nuclear and cytoplasmic staining for IRF-1 was evaluated using almost the same method as that for γH2AX described above. The staining proportion of IRF-1 was scored as follows: 0, no staining; 1+, 1–25%; 2+, 26–75%; and 3+, 76–100%. The final score was defined as the ratio of the score multiplied by the intensity score (0, 1+, 2+, 3+, 4+, 6+, or 9+). IRF-1 immunoreactivity was scored as 0–4+ or 6–9+, defined as low or high expression, respectively. To evaluate the infiltrating immune cells, the total number of infiltrating CD8, CD68, CD163, and FOXP3 positive cells in five high-power fields was counted in the sections. MMRD was defined as the complete absence of expression of at least one mismatch repair protein [43] (MLH1, MSH2, MSH6, or PMS2).

### 4.4. Multicolor Immunofluorescence Staining for HLA-I and CD8

The sections were prepared, and endogenous peroxidase was blocked as described above. Nonspecific binding sites were blocked by incubation with Protein Block Serum-Free Reagent for 30 min, and the sections were incubated overnight at 4 °C with the primary antibodies against HLA Class I-A, B, C (Hokudo, Sapporo, Japan, EMR8-5, mouse mAb, 1:400 dilution) and CD8 (Abcam, Cambridge, MA, USA, ab4055, rabbit pAb, 1:400 dilution). Multiplex covalent labeling (HLA-I, Opal 520 Fluorophore, OP-001001; CD8, Opal 570 Fluorophore, OP-001003) with tyramide signal amplification (Akoya Biosciences, MA, USA) was performed according to the manufacturer’s protocol. All sections were counterstained with DAPI and examined under an All-in-One BZ-X710 fluorescence microscope (KEYENCE Corporation, Osaka, Japan).

### 4.5. Statistical Analyses

The JMP Pro 14.0 software package (SAS Institute Inc., Cary, NC, USA) was used for all the statistical analyses. Statistical analysis was performed using Wilcoxon and Steel-Dwass tests. Differences were considered statistically significant at *p* < 0.05.

## Figures and Tables

**Figure 1 ijms-24-13648-f001:**
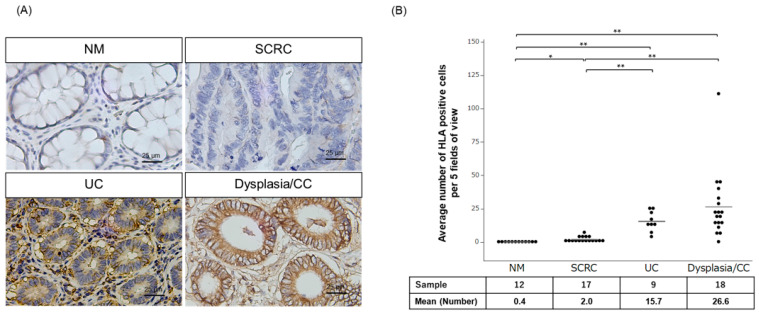
Immunohistochemical staining of HLA-I in NM, SCRC, UC, and dysplasia/CC. (**A**) Upper panel shows the HLA-I expression in NM and SCRC. The lower panel shows the membrane expression of HLA-I in UC and dysplasia/CC. Scale bar, 25 µm. (Original magnification, ×40). (**B**) The Average number of membrane HLA-I-positive epithelial cells in NM (n = 12), SCRC (n = 17), UC (n = 9), dysplasia (n = 5), and CC (n = 13) tissues. The dots indicate the average number of HLA-I positive cells in the five fields of view in the sample. * *p* < 0.05; ** *p* < 0.01. NM, normal mucosa adjacent to SCRC; SCRC, sporadic colorectal cancer; UC, ulcerative colitis; CC, colitis cancer.

**Figure 2 ijms-24-13648-f002:**
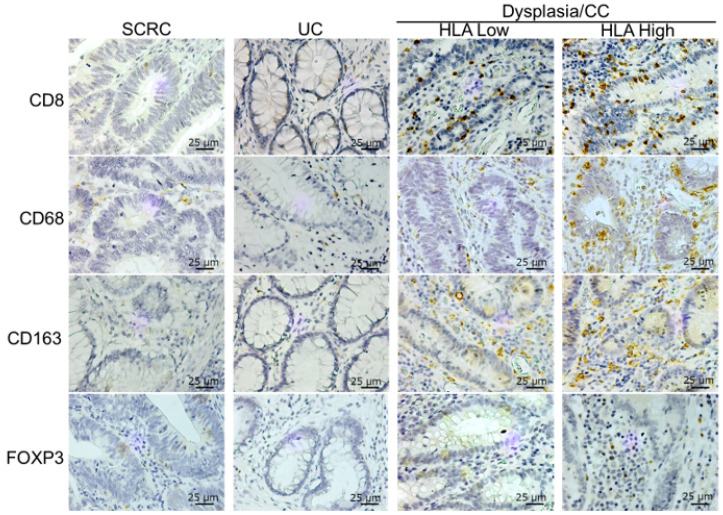
Immunohistochemical staining of CD8, CD68, CD163, and FOXP3 in tissues from SCRC, UC, and dysplasia/CC with low and high HLA-I. Representative immunohistochemical images of immune cell markers such as CD8 (cytotoxic T cell marker), CD68 (M1-like macrophage marker), CD163 (M2-like macrophage marker), and FOXP3 (regulatory T cell marker) in tissues from SCRC, UC, and dysplasia/CC with low and high HLA-I. Scale bar, 25 μm. (original magnification, ×40). SCRC, sporadic colorectal cancer; UC, ulcerative colitis; CC, colitis cancer.

**Figure 3 ijms-24-13648-f003:**
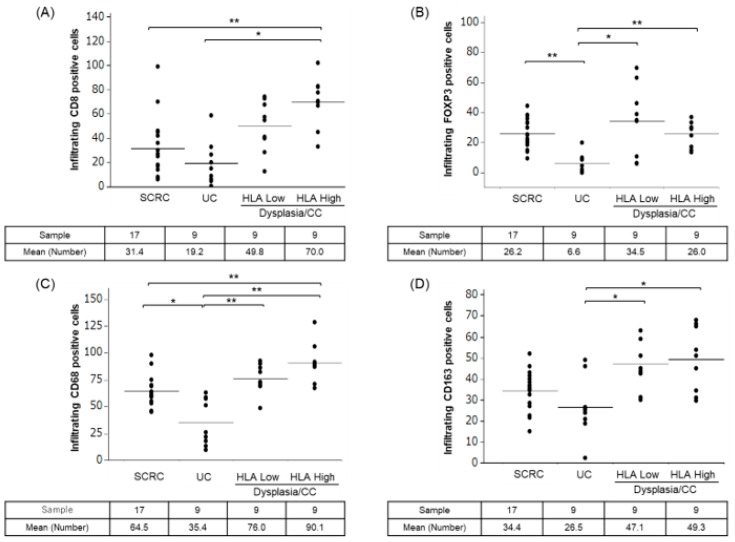
Comparison of infiltrating immune cell counts between SCRC, UC, and dysplasia/CC with low and high HLA-I. (**A**) The infiltrating cell number of CD8 (cytotoxic T cell marker) positive cells in SCRC, UC, and dysplasia/CC with low and high HLA-I. (**B**) The infiltrating cell number of FOXP3 (regulatory T cell marker) positive cells in SCRC, UC, and dysplasia/CC with low and high HLA-I. (**C**) The infiltrating cell number of CD68 (M1-like pro-inflammatory macrophage marker) positive cells in SCRC, UC, and dysplasia/CC with low and high HLA-I. (**D**) The infiltrating cell number of CD163 (M2-like anti-inflammatory macrophage marker) positive cells in SCRC, UC, and dysplasia/CC with low and high HLA-I. * *p* < 0.05; ** *p* < 0.01. SCRC, sporadic colorectal cancer; UC, ulcerative colitis; CC, colitis.

**Figure 4 ijms-24-13648-f004:**
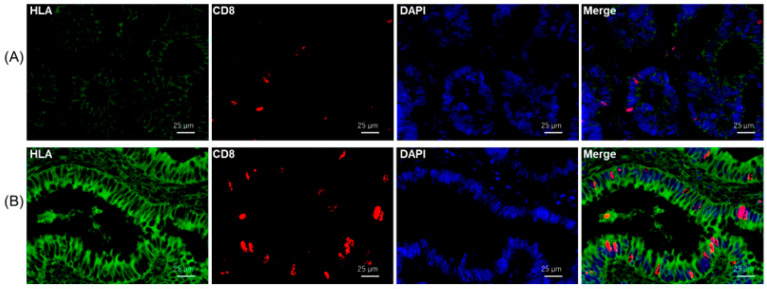
Immunofluorescence analysis for HLA-I and CD8 expression in dysplasia/CC tissues. Representative dysplasia/CC tissues with low (**A**) or high (**B**) HLA-I expression were immunostained with anti-HLA-I (green) and anti-CD8 (red) antibodies. All sections were counterstained with 4′,6-diamidino-2-phenylindole (DAPI) (blue). Scale bar = 25 μm (original magnification, ×40).

**Table 1 ijms-24-13648-t001:** Clinicopathological characteristics in 9 patients with Dysplasia/CC.

Case Number	Age	Sex	Pathological Diagnosis	MMRD	Location	Differentiation	T Factor	N Factor	M Factor	Stage	Average Number of HLA Positive Cells per 5 Fields of View	HLA	γH2AX	PD-L1	IRF-1
Case1	72	Female	UC	−							4.2	Low	−	+	+
			CC	−	S	Poor	4b	0	0	IIC	20	Low	+	+	+
Case2	62	Male	UC	−							22.2	Low	−	+	+
			CC	−	Rb	Well	2	0	0	I	22.6	High	+	+	+
			CC	−	D	Well	1	0	0	I	0.4	High	+	+	+
Case3	37	Male	UC	−							13.6	Low	−	+	+
			Dysplasia	−	A						111.2	Low	+	+	+
			Dysplasia	−	A						6.8	High	+	+	+
			Dysplasia	−	Ce						45.4	High	+	+	+
			CC	−	A	Moderate	4a	1a	0	IIIB	40.2	High	+	+	+
			CC	−	S	Moderate	2	0	0	IIIB	22.6	High	+	+	+
			CC	−	Ce	Moderate	3	0	0	IIIB	33	High	+	+	+
Case4	38	Male	UC	−							12.6	Low	−	+	+
			CC	−	S	Moderate	4a	0	0	IIB	28.9	High	+	+	+
Case5	51	Female	UC	−							24.8	High	−	−	+
			CC	+	Ra	Well	Tis	0	0	0	45	High	+	+	+
Case6	42	Female	UC	−							6.6	Low	−	+	+
			Dysplasia	−	S						15	Low	+	+	+
			CC	−	S	Well	2	0	0	I	18.6	Low	+	+	+
Case7	55	Male	UC	−							17.2	Low	−	+	+
			Dysplasia	−	S						6.8	Low	+	+	+
			CC	−	S	Well	4a	2a	M1b	IVB	15.2	Low	+	+	+
Case8	51	Male	UC	−							26	Low	−	−	+
			CC	−	D	Well	Tis	0	0	0	22.3	High	+	+	+
			CC	−	D	Well	Tis	0	0	0	11.2	High	+	+	+
Case9	38	Male	UC	−							14	Low	−	+	+
			CC	+	Rb	Well	T1b	0	0	I	13.8	Low	+	+	+

CC, colitic cancer; UC, Ulcerative Colitis; MMRD, mismatch repair deficiency; Ce, cecum; A, ascending colon; D, descending colon; S, sigmoid colon; Ra, above rectum; Rb, below rectum. + , Positive or high expression of target proteins; − , Negative or low expression of target proteins.

**Table 2 ijms-24-13648-t002:** Clinicopathological characteristics in 17 sporadic colon cancer patients.

Case Number	Age	Sex	Pathological Diagnosis	MMRD	Location	Differentiation	T Factor	N Factor	M Factor	Stage	Average Number of HLA Positive Cells per 5 Fields of View	HLA	γH2AX	PD-L1	IRF-1
Case1	72	Female	UC	−							4.2	Low	−	+	+
			CC	−	S	Poor	4b	0	0	IIC	20	Low	+	+	+
Case2	62	Male	UC	−							22.2	Low	−	+	+
			CC	−	Rb	Well	2	0	0	I	22.6	High	+	+	+
			CC	−	D	Well	1	0	0	I	0.4	High	+	+	+
Case3	37	Male	UC	−							13.6	Low	−	+	+
			Dysplasia	−	A						111.2	Low	+	+	+
			Dysplasia	−	A						6.8	High	+	+	+
			Dysplasia	−	Ce						45.4	High	+	+	+
			CC	−	A	Moderate	4a	1a	0	IIIB	40.2	High	+	+	+
			CC	−	S	Moderate	2	0	0	IIIB	22.6	High	+	+	+
			CC	−	Ce	Moderate	3	0	0	IIIB	33	High	+	+	+
Case4	38	Male	UC	−							12.6	Low	−	+	+
			CC	−	S	Moderate	4a	0	0	IIB	28.9	High	+	+	+
Case5	51	Female	UC	−							24.8	High	−	−	+
			CC	+	Ra	Well	Tis	0	0	0	45	High	+	+	+
Case6	42	Female	UC	−							6.6	Low	−	+	+
			Dysplasia	−	S						15	Low	+	+	+
			CC	−	S	Well	2	0	0	I	18.6	Low	+	+	+
Case7	55	Male	UC	−							17.2	Low	−	+	+
			Dysplasia	−	S						6.8	Low	+	+	+
			CC	−	S	Well	4a	2a	M1b	IVB	15.2	Low	+	+	+
Case8	51	Male	UC	−							26	Low	−	−	+
			CC	−	D	Well	Tis	0	0	0	22.3	High	+	+	+
			CC	−	D	Well	Tis	0	0	0	11.2	High	+	+	+
Case9	38	Male	UC	−							14	Low	−	+	+
			CC	+	Rb	Well	T1b	0	0	I	13.8	Low	+	+	+

MMRD, mismatch repair deficiency; Ce, cecum; A, ascending colon; T, transverse colon; S, sigmoid colon; Ra, above the rectum. + , Positive or high expression of target proteins; − , Negative or low expression of target proteins.

## Data Availability

The data supporting the findings of this study are available from the corresponding author upon reasonable request.

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
