# Peer review of "HLA Class I Expression Is Associated with DNA Damage and Immune Cell Infiltration into Dysplastic and Neoplastic Lesions in Ulcerative Colitis"

_ijms, 2023, doi:10.3390/ijms241713648_

Round 1
Reviewer 1 Report
A significant increase in the number of HLA-I-positive epithelial cells in UC and dysplasia/CC tissues compared to NM and SCRC and the association between HLA I expression and the high expression of the DNA damage marker γH2AX was observed in dysplasia/CC. Indeed, the immune checkpoint protein PD-L1 was observed. Moreover, the infiltration of CD8-positive T cells and CD68-positive macrophages in dysplasia/CC with high HLA-I levels was significantly higher than that in UC and SCRC.
This is what the authors discussed and demonstrated in their manuscript.
In general I think that this is a good and interesting work, even if there are some limitations that also the authors underlie in the conclusion part.
Indeed, there are some parts in the text, mostly in the introduction, that could be improved.
The quality of English language is good. No issue I detected.
Even if the number of patients enrolled is low, in general the experiments performed are well described in the result section and the figures and table are well done.
The authors could find below all the comments I did divided for all section:
Introduction:
I think that the authors could improve this part, for examples by adding more the role of HLA molecules, in particular HLA class I that is mentioned in the manuscript. Indeed, the relationship between HLA molecules and UC or more in general inflammatory bowel diseases could be discussed in order to clearly introduce the importance of these molecules during a inflammatory or neoplastic process. Please revise the introduction part, I really think that this could support better the data the authors produced and discussed here.
The correlation between DNA damage and HLA I expression is not so clear. Please the authors should add more details. How DNA damage could stimulate HLA I presentation? Whose cells are involved in the HLA molecules presentation? (Lines 55- 59). The authors should decide to include or not HLA class II molecule.
Lines 67-69: Please formulate better this part: “...the importance of DNA damage...” Please the authors should write better this part thus trying to make a sense on the real correlation between DNA damage, HLA, PDL-1.
Lines 69-75: I think that this part could be shift in another section (results,discussion) or if the authors want to maintain in the Introduction section I think that the authors should rewrite better this section trying to make sense to all the section.
Lines 67-75: Please try to develop better this final part of the introduction underlie the aim of this study.
Results:
Lines 81-83: The authors assessed that the expression of HLA class I on target cells indicate the antigen presentation by HLA class I but other experiments could be performed in order to demonstrate the HLA presenting process. The only expression of these molecules could be associated with a positive signaling for adaptive immune cells (CD8 T cells),considering that HLA I molecules contribute to the presentation of antigen peptide from an APC. Please this part should be revised, if the authors want to demonstrate the antigen presentation process by HLA molecules they should do other experiments otherwise, I suggest just to explain in another way these results.
The Discussion is well written and the conclusion is introducing a possible therapeutic strategy to prevent DNA damage and regulate immune cells processing.

Author Response
Our point-by-point responses to the reviewers’ comments and suggestions are listed below:
We thank you for taking the time and effort necessary to review our manuscript and provide us with these valuable comments and suggestions. Accordingly, we revised our manuscript and made changes to it. Please note that changes to the manuscript are highlighted in red font for your convenience.
Reviewer 1
Query 1.
I think that the authors could improve this part, for examples by adding more the role of HLA molecules, in particular HLA class I that is mentioned in the manuscript. Indeed, the relationship between HLA molecules and UC or more in general inflammatory bowel diseases could be discussed in order to clearly introduce the importance of these molecules during a inflammatory or neoplastic process. Please revise the introduction part, I really think that this could support better the data the authors produced and discussed here.
Response 1.
We thank you for this important suggestion. Accordingly, we revised this part and added more sentences and references to the Introduction section as follows:
" Non-steroidal anti-inflammatory drugs, such as aspirin, have been reported to prevent sporadic colorectal cancer (SCRC) by suppressing chronic inflammation in the colorectal mucosa [1-3]. However, chronic colon inflammation due to autoimmune responses could promote ulcerative colitis (UC) and rare colitis-associated dysplasia/colitic cancer (CC) via aggressive infiltration of inflammatory lymphocytes and macrophages [4,5]. The human leukocyte antigen (HLA) molecule is implicated in the pathogenesis of autoimmune chronic inflammation of the colon. The highly polymorphic classical class I and II HLA genes located at chromosome 6p21.3 are important for lymphocyte and immunoregulatory functions in patients with inflammatory bowel diseases, including UC [6]. Many researchers have reported the importance of HLA class II genotypes such as HLA-DRB1*0103, HLA-DRB1*1502, and HLA-DRB1*04 in relating to UC susceptibility. On the other hand, the significant expression of HLA class I (HLA-I) on cell surface, which is important for presenting antigen to cytotoxic T cells, has been reported in patients with several cancer types [7-10]. Regarding SCRC, HLA-I expression was lost in 30–73% of patients with tumors [11,12]: suggesting that lack of HLA-I expression may cause both initial resistance and secondary immune escape after existing immune checkpoint inhibitors due to cytotoxic T cell activation [13-15]. However, few studies assessed the significance of HLA-I expression in patients with relatively rare UC and dysplasia/CC. (Lines: 42-60)"
Query 2.
The correlation between DNA damage and HLA I expression is not so clear. Please the authors should add more details. How DNA damage could stimulate HLA I presentation? Whose cells are involved in the HLA molecules presentation? (Lines 55- 59). The authors should decide to include or not HLA class II molecule.
Response 2.
We thank you for this insightful suggestion. We revised the following paragraphs regarding the HLA and DNA damage as follow:
In the introduction section:
" Regarding SCRC, HLA-I expression was lost in 30–73% of patients with tumors [11,12]: suggesting that lack of HLA-I expression in SCRC may cause both initial resistance and secondary immune escape after existing immune checkpoint inhibitors due to cytotoxic T cell activation [13-15]. However, few studies assessed the significance of HLA-I expression in patients with relatively rare UC and dysplasia/CC. Therefore, we focused on assessing the relationship between immune cell infiltration and HLA-I expression, not HLA-class II, on the epithelial cells in SCRC and dysplasia/CC tissues in this study. (lines: 55-62)"
"Uchihara et al. have recently described a novel regulation mechanism of HLA-I expression by which antigen transport via TAP1/2 could promote presentation of HLA-I to the plasma membrane regardless of DNA damage-inducing methods, such as radiation or anticancer drugs [16]. DNA damage-induced HLA-I presentation could be predicted to activate signaling downstream of T-cell immunity. In contrast, Sato et al. reported that expression of the immune checkpoint protein: PD-L1 is upregulated by DNA damage response [17]. These reports indicated that DNA damage could induce two opposing events: inflammatory activation via antigen presentation by HLA-I and immune tolerance by induction of the immune checkpoint protein: PD-L1. Concerning the relationship among DNA damage, PD-L1 expression, and immune cell infiltration in dysplasia/CC tissues, our group reported that CD8-positive cytotoxic T-cell infiltration was higher in PD-L1 expressing dysplasia/CC tissues with DNA damage than in those with SCRC without DNA damage [18]. However, the effects of HLA-I on immune cell infiltration in dysplasia/CC tissues with DNA damage and PD-L1 expression have not been thoroughly evaluated. (Lines: 63-77)"
Query 3.
Lines 67-69: Please formulate better this part: “...the importance of DNA damage...” Please the authors should write better this part thus trying to make a sense on the real correlation between DNA damage, HLA, PDL-1.
Response 3.
We thank you for this suggestion. Accordingly, we revised the purpose of this study in the Introduction section as follows:
"This study aimed to compare the association between HLA-I expression levels and immune cell infiltration in dysplasia/CC tissues, which we previously reported to accumulate DNA damage and PD-L1, with tissues from SCRC samples. (Lines: 77-79)"
Query 4.
Lines 69-75: I think that this part could be shift in another section (results, discussion) or if the authors want to maintain in the Introduction section, I think that the authors should rewrite better this section trying to make sense to all the section.
Response 4.
We thank you for this suggestion. According to this suggestion, we moved the sentences to the first paragraph of the discussion part as follows:
" In this study, we performed immunohistochemical staining for HLA-I, PD-L1, γH2AX (DNA damage marker), IRF-1 (a representative HLA inducible factor), and immune cell markers such as CD8 (cytotoxic T cell marker), FOXP3 (regulatory T cell marker), CD68 (M1-like pro-inflammatory macrophage marker), and CD163 (M2-like anti-inflammatory macrophage marker) in surgically resected specimens from 17 patients with SCRC with 12 adjacent normal mucosa (NM) and 9 patients with UC with 18 dysplasia/CC tumors. Our results demonstrated a significant increase in the number of HLA-I-positive epithelial cells in UC and dysplasia/CC tissues compared to NM and SCRC. In addition, the high HLA-I expression in dysplasia/CC tissues was associated with high expression of the DNA damage marker (γH2AX) and a representative immune checkpoint protein (PD-L1) compared to SCRC tissues without high HLA-I. Moreover, the infiltration of CD8-positive T cells and CD68-positive macrophages in dysplasia/CC tissues with high HLA-I levels was significantly higher than that in UC and SCRC tissues. (Lines: 159-172)"
Query 5.
Lines 81-83: The authors assessed that the expression of HLA class I on target cells indicate the antigen presentation by HLA class I but other experiments could be performed in order to demonstrate the HLA presenting process. The only expression of these molecules could be associated with a positive signaling for adaptive immune cells (CD8 T cells), considering that HLA I molecules contribute to the presentation of antigen peptide from an APC. Please this part should be revised, if the authors want to demonstrate the antigen presentation process by HLA molecules they should do other experiments otherwise, I suggest just to explain in another way these results.
Response 5.
We thank you for this suggestion. Accordingly, we revised the description about the potential of antigen presentation by membrane HLA-I expression in the Results section as follows:
"HLA-I expression was predominantly detected in the membrane of the epithelial cells in tissues from UC and CC samples (Figure 1A). (Lines: 85-86) "
Reviewer 2 Report
The authors link HLA-I expression in various immune cell types with inflammation and DNA damage as well as PDL1 expression. The comparison of UC, CC and SCRC in this context is novel and relevant in the context of stratifying patients for ICB. The paper is informative, the data convincing and well presented, however it would be very helpful to know if the levels of HLA, gammaH2AX and PDL1 correlate in any way with patient outcome within each group. In other words, is this higher level of inflammation associated with HLA expression beneficial or detrimental once the cancer has developed (in the CC group)?
Author Response
Reviewer 2.
Comment 1.
The authors link HLA-I expression in various immune cell types with inflammation and DNA damage as well as PDL1 expression. The comparison of UC, CC and SCRC in this context is novel and relevant in the context of stratifying patients for ICB. The paper is informative, the data convincing and well presented, however it would be very helpful to know if the levels of HLA, gammaH2AX and PDL1 correlate in any way with patient outcome within each group. In other words, is this higher level of inflammation associated with HLA expression beneficial or detrimental once the cancer has developed (in the CC group)?
Response to the comment.
We thank you for the positive comments.
Increased-immune cell infiltration has been associated with a favorable prognosis in patients with several cancer types. In addition, as mentioned in the Discussion section that the sensitivity to immune checkpoint inhibitors has been reported to be associated with high levels of immune cell infiltration. On the other hand, PD-L1 and gammaH2AX have been reported to correlate with cancer aggressiveness and poor prognosis. Therefore, further studies are warranted to clarify the clinical significance of HLA expression including a large cohort with CC.
Round 2
Reviewer 2 Report
While the authors did not directly answer the question, if the issue of inflammation versus DNA damage signature and prognosis and ICB response is covered in the discussion, it is acceptable.